# Donor-Specific Cell-Free DNA qPCR Quantification as a Noninvasive Accurate Biomarker for Early Rejection Detection in Liver Transplantation

**DOI:** 10.3390/jcm12010036

**Published:** 2022-12-21

**Authors:** Noelia García-Fernández, Hada C. Macher, Gonzalo Suárez-Artacho, Miguel Ángel Gómez-Bravo, Patrocinio Molinero, Juan Miguel Guerrero, Manuel Porras-López, Amalia Rubio

**Affiliations:** 1Department of Clinical Biochemistry, Instituto de Investigaciones Biomédicas de Sevilla, IBIS (University of Seville, HUVR, Junta de Andalucía, CSIC), 41013 Seville, Spain; 2Hepatobiliary and Liver Transplantation Unit, Virgen del Rocío University Hospital, 41013 Seville, Spain; 3Department of Medical Biochemistry and Molecular Biology and Immunology, Instituto de Investigaciones Biomédicas de Sevilla, IBIS (University of Seville, HUVR, Junta de Andalucía, CSIC), 41013 Seville, Spain; 4Intensive Care Unit, Virgen del Rocio University Hospital, 41013 Seville, Spain

**Keywords:** donor-specific cfDNA, liver transplantation, graft rejection, noninvasive biomarker

## Abstract

(1) Background: Graft-cell-free DNA (cfDNA) in the circulation of liver transplant recipients has been proposed as a noninvasive biomarker of organ rejection. The aim of this study was to detect donor-specific cfDNA (ds-cfDNA) in the recipient’s serum after either liver damage or rejection using a qPCR-based method. (2) Methods: We proposed a qPCR method based on the amplification of 10 specific insertion–deletion (InDel) polymorphisms to detect donor-specific circulating DNA diluted in the recipient cfDNA. ds-cfDNA from 67 patients was evaluated during the first month post-transplantation. (3) Results: Graft rejection in the first month post-transplantation was reported in 13 patients. Patients without liver complications showed a transitory increase in ds-cfDNA levels at transplantation. Patients with rejection showed significant differences in ds-cfDNA increase over basal levels at both the rejection time point and several days before rejection. Receiver operator characteristic (ROC) analysis showed that ds-cfDNA levels discriminated rejection, with an AUC of 0.96. Maximizing both sensitivity and specificity, a threshold cutoff of 8.6% provided an estimated positive and negative predictive value of 99% and 60%, respectively. (4) Conclusions: These results suggest that ds-cfDNA may be a useful marker of graft integrity in liver transplant patients to screen for rejection and liver damage.

## 1. Introduction

The early detection of graft damage is a major clinical concern in solid organ transplantation (TX). After liver transplantation, biopsy is considered the gold standard technique for evaluating organ rejection. However, this procedure is invasive and can result in clinical complications. Liver biopsy provides diagnostic confirmation of many graft complications, although it involves risk of morbidity (1%) and mortality (from 0.1 to 0.01%) to the patient [1]. To adjust the treatment, continued assessment of graft integrity is essential for monitoring liver transplant patients. However, the performance of frequent serial liver biopsies should not be considered a possible approach, mainly because of an increase in patient risk exposure. Moreover, this is a costly and time-consuming procedure that is inconvenient for quick changes in medication treatment.

A new approach has gained importance in recent years with the detection of a noninvasive biomarker capable of rapidly detecting organ rejection: cell-free DNA (cfDNA) derived from graft cells. Thus, graft cfDNA in the circulation of liver transplant recipients has been proposed as a potential biomarker for organ rejection or cellular graft injury. The first evidence dates from 1998, when Lo et al. described the presence of chromosome Y-specific sequences in plasma obtained from a female host who received a kidney from a male donor [2]. They proposed that the DNA released from necrotic or apoptotic cells during rejection may be a useful marker of graft damage. Since then, several groups have determined graft cfDNA using different techniques, such as whole-genome sequencing [3,4,5,6], digital PCR [7,8,9], and real-time PCR [10,11,12].

Real-time PCR (qPCR) of specific organ cfDNA has been proven to be a noninvasive and advantageous approach for monitoring either liver damage or rejection after TX in a specific subgroup of patients, such as a female host receiving an organ from a male donor [10,11]. To expand this methodology to the entire population, we proposed, in a previous work, a method based on a set of insertion–deletion (InDel) diallelic polymorphisms that are already described in the general population [13]. With this approach, we were able to detect increases in donor-specific cfDNA (ds-cfDNA) in the recipient’s serum after either heart damage or heart transplantation rejection in a patient cohort.

The aim of this study was to detect ds-cfDNA present in the recipient’s serum after either liver damage or rejection using a qPCR-based method. For the proposed approach, the detection of a donor-recipient DNA mismatch for InDel sequences is key for monitoring organ health after TX. Thus, we first determined the donor–recipient DNA mismatch in liver transplant patients. We considered an informative mismatch in the amplification of the InDel sequence in the donor but not in the recipient. Thereafter, we monitored ds-cfDNA levels in the host serum during the first month of follow-up. We described a significant increase in ds-cfDNA related to liver damage on the day of rejection diagnosis and several days before. Moreover, the high negative predictive value obtained in this study suggests that ds-cfDNA quantification using qPCR may be helpful in deciding on the indication for biopsy in the setting of elevated liver biomarkers.

## 2. Materials and Methods

### 2.1. Study Subjects

All consecutive patients who underwent orthotropic liver TX at Virgen del Rocio University Hospital in Seville (Spain) during a 4-year period were offered inclusion in the study for the monitoring of ds-cfDNA during the first month. Patients were clinically evaluated at least 2 years after inclusion. After placement on the waiting list, the patients were informed about the study (unless they were in such a poor clinical status that precluded the individual from understanding the rationale behind the study). After agreement, the patients provided signed informed consent, and they were included in the study. Patients who underwent orthotropic liver TX fulfilled the following inclusion criteria: (a) be over 18 years old and (b) sign the informed consent.

During surgery, samples from both donors and recipients were collected. A blood sample from the central line was collected at two different times during surgery: a basal sample at the anhepatic phase and 15 min after graft reperfusion. When a recipient tissue sample was not available, a basal EDTA blood sample was used for obtaining DNA from peripheral blood mononuclear cells.

Patients from whom either a donor or a recipient tissue or blood sample could not be collected were excluded from the study. Cases with either absent or invalid serum samples at the moment of TX or with less than 1 week of follow-up were also excluded from the study. Finally, 97 liver-transplanted patients were included. The general characteristics of the patients are shown in Table 1.

### 2.2. Patient Surveillance and Treatment

Patients were evaluated clinically and biologically both during their stay in the intensive care unit (ICU) and the ward. Blood samples were drawn by peripheral venipuncture daily during the first three days, and every 2–3 days thereafter in a Vacuette^®^ 9 mL Z Serum Separator Clot Activator (Greiner Bio-One, Kremsmünster, Austria). We monitored the cfDNA in the serum and the standard biochemical, hematological, and coagulation parameters as well as determined the immunosuppressive drug levels. For cfDNA determination, 10 mL of blood samples were drawn and centrifuged 5 min at 3500 rpm at room temperature within 6 h after extraction, and the serum was aliquoted and frozen at −80 °C for future determination. After the patients were discharged, they were followed up at scheduled visits, drawing a blood sample at every visit. DNA from frozen samples was obtained and stored for no longer than 6 months.

Immunosuppressive treatment for cohort patients consisted of Tacrolimus, mofetil mycophenolate, and steroids. A 7 ng/mL blood level of Calcineurin inhibitors was required during the first month, between 5–7 ng/mL during the first year and less than 5 ng/mL after the first year. With a protocol based on Basiliximab, tacrolimus delayed reduced dose, and mofetil mycophenolate and steroids were used in patients with chronic renal dysfunction.

### 2.3. Ethics Statement

Ethical approval for the study protocol was granted by the Medical Research Ethics Board of the Virgen del Rocío University Hospital of Seville. The clinical investigation was conducted according to the principles expressed in the Declaration of Helsinki.

### 2.4. General Workflow

We propose an approach to detect ds-cfDNA in the recipient based on the existence of long genomic sequences (InDels) which are either present or absent in the individual genotype. We have previously described a panel of 10 InDels that were useful in transplant monitoring to differentiate donor cfDNA from that of the recipient [8,13]. The panel consists of four null alleles (GSTM1, GSTT1, SRY and RhD) and six InDels (DCP1, Xq28, R271, rs4399, FVII, and THYR). The chromosome position, length of the insertion sequence and frequency of the different InDels is previously described [8].

The workflow for graft injury evaluation involves two steps, as follows (Figure 1). The first step comprised InDel identification. Thus, tissue samples from both donors and recipients were analyzed to determine a donor–recipient mismatch for the InDels analyzed. A mismatch was considered informative when a deleted sequence on the recipient DNA, but not in the donor DNA, was observed. The second step consists of the quantification of ds-cfDNA in the recipient’s serum. After the detection of an informative mismatch, cfDNA from sequential serum samples was analyzed by qPCR. The presence of elevated ds-cfDNA circulating in serum from the recipient is indicative of any type of transplanted liver complication. We also quantified beta-globin gene values as a control for general damage or patient clinical worsening.

### 2.5. DNA Extraction from Tissue and Serum Samples

DNA from tissue samples was extracted using QIAcube (Qiagen GmbH, Hilden, Germany), according to the manufacturer’s protocol; 5 milligrams of donor and recipient biopsy samples were previously minced and incubated with 180 µL of ATL lysis buffer (animal tissue lysis buffer) and 20 µL of proteinase K for 2 hours at 56 °C. The DNA was eluted in a final volume of 100 μL and was frozen at −80 °C until determination.

DNA from 400 μL of serum samples was extracted using the automatized MagNaPure Compact Instrument (Roche Diagnostics, Basel, Switzerland) using the Magna Pure Compact Nucleic Acid Isolation Kit I, according to the protocol “Total NA Plasma 100 400 V3 1”. The DNA was eluted in a final volume of 50 μL and was frozen at −80 °C until either qPCR.

Quantification of the nucleic acids after DNA isolation was performed using Qubit 3.0 fluorometry (Thermo Fisher Scientific, Waltham, MA, USA) according to the manufacturer’s instructions.

### 2.6. Determination of Donor–Recipient Mismatch and Monitoring Organ Specific cfDNA

Selected InDels were amplified from donor and recipient tissue DNA by qPCR assay using the Light-Cycler 480 Real-Time PCR instrument (Roche Diagnostics, Basel, Switzerland). We followed defined guidelines for qPCR analyisis [14].

Briefly, 2 microliters of DNA were amplified in a final volume of 20 μL containing 200 nM of primers and 100 nM of the probe using the LC480 Probes Master Kit (Roche Diagnostics, Basel, Switzerland). The 2× concentrated master mix is optimized for a fixed MgCl_2_ concentration, which works with nearly all primer combinations. No adjustment in the MgCl_2_ concentration is needed to amplify different sequences. qPCR was performed at 95 °C for 5″ and at the specific InDel Tm for 20″ for 40 cycles. The standards for the calibration curve were based on the dilutions of human genomic DNA (Roche Diagnostics, Basel, Switzerland). Calibration curve slopes ranged from 3.34 to 3.85 and provided a PCR efficiency range close to 2 [1.81–1.99]. The lower standard provided a limit of detection of 1.5 genomic equivalents (GE)/mL serum. The error value (mean squared error of the single data points fit to the regression line), a measure of the accuracy of the quantification result, was always <0.2 [0.0004–0.01]. The primer and probe sequences, fragment length as well as annealing temperature have been previously described [8].

For quantification of donor DNA in the recipient’s serum 5 microliters of DNA was amplified by qPCR assay using the Light-Cycler 480 Real-Time PCR instrument (Roche Diagnostics, Basel, Switzerland) following the same protocol as described above.

DNA from each patient was assayed in one round, and all samples were assayed in duplicate. No-template control was included to asses any possible contamination problem in the assay. The final concentration was calculated according to the following formula:$$c=GE\times \frac{V_{DNA}}{V_{PCR}}\times \frac{1}{V_{ext}}$$
where *GE* represents a genomic equivalent, *V_DNA_* represents the total volume of cfDNA obtained after extraction from serum, *V_PCR_* represents the sample volume used for PCR, and *V_ext_* represents the volume of extracted serum. A conversion factor of 6.6 pg of DNA per diploid cell was used to express the cfDNA concentration as genome equivalents.

### 2.7. Statistical Analysis

All statistical analysis was performed with IBM SPSS version 26 (IBM, Armonk, NY, USA). Continuous data are presented by median and range. Comparisons between two continuous variables were performed by *t*-test for normal distribution. Mann–Whitney U-test was used to compare differences between two independent groups when the dependent variable was not normally distributed. Multiple comparison analysis was performed by ANOVA test and post hoc analysis by Bonferroni test.

Receiver operator characteristic (ROC) analyses were performed to determine the best parameter to discriminate between biopsy-proven rejection (BPR) and stable samples. ROC curves were performed for classical hepatic markers and graft cfDNA increase in the same analysis. Samples were only included when a determination of all parameters studied was available. Threshold values for discrimination of BPR and stable samples were calculated based on the maximum Youden index. Based in the threshold value obtained, both the positive predictive value (PPV) and the negative predictive value (NPV) were calculated.

## 3. Results

### 3.1. Characteristic of the Patients

The general characteristics of the patients (total, stable, and rejection) are shown in Table 1. We defined stable patients as patients that did not suffered rejection but may suffer any kind of post TX complication. The patients’ age ranged from 18 to 71 years old. Ten patients stayed in the ICU for several days prior to TX, although only two patients spent more than 4 days in the ICU (34 and 58 days). After surgery, all patients were admitted to the ICU for durations (3–54 days) depending on the general outcome of the patient. Nine patients (10.3%) died after TX during the period studied. One patient died early after acute graft rejection at day 420, and the others died due to different causes over a wide period of time ranging from 4 days to 34 months after TX. The causes of death are shown in the Appendix A. Two of these patients were not evaluated during the follow-up because we did not find a valid informative InDel.

Graft rejection was reported in 20 patients (20.6%), and in most cases, it was controlled by increasing the immunosuppressant dose. Of the 20 patients, 13 (13.4%) experienced rejection during the first month of follow up. In two cases, a serum sample was not available at the time of rejection for cfDNA determination.

Table 1 also shows the comparison between patients with and without rejection during the first month after TX. Both age and total hospitalization time seemed to differ between patients in both groups. The median age was significantly higher in the stable patient group, and a significantly longer hospitalization time was observed in patients with rejection. Both the warm and cold ischemia times were not significantly different between the groups. The ischemia time was always within the optimal range for organ preservation. We observed a warm ischemia median of 30 min for both groups, with most patients (75%) under 36 min, and only two patients presented a warm ischemia time of 1 h. The median cold ischemia time was 6 h, and only one patient received a liver after 9 h of cold ischemia.

To obtain a more balanced population between both the groups, we also compared fully stable patients (77%) and patients with any kind of hepatic damage during the first month of follow up (23%). Similar results were found for most variables, although in this comparison, differences were more significant for hospitalization time and less evident for patient age. In addition, a higher proportion of patients with hepatic damage stayed in the ICU during their previous TX (Appendix A).

### 3.2. Graft cfDNA Monitoring during One Month after TX

An informative mismatch was observed in 69 (72%) patients; therefore, this was the study population during the follow up. Of these 69 patients, 13 experienced organ rejection during the first month after TX. Figure 2 shows an expected high increase in specific genomic marker levels during the first 24 h after TX, which diminished in different patterns depending on patient evolution. As illustrated in Figure 2A, patients without complications during the first month with good evolution (open circles) or bad evolution (black circles) during the 2 years after TX showed rapid decrease in graft cfDNA and presented basal levels during this period. Although a slight difference was observed between the evolution of these two groups of patients, the difference was not statistically significant. Figure 2B shows the evolution of patients with rejection. An increase in the percentage of graft cfDNA, indicated by arrows, was observed on days when one or more patients experienced BPR.

Beta-globin cfDNA was also measured as a control for general damage or clinical worsening. As for ds-cfDNA levels, we observed that total cfDNA levels increased at the moment of organ reperfusion, which then diminished during the follow-up. Total cfDNA levels were quite variable and did not reach basal levels during the first month after TX (Appendix A). Functional liver marker values for stable patients were quite variable and in many cases did not reach basal levels during the first month of follow up (Appendix A).

### 3.3. Changes in Percentage of Graft cfDNA and Classical Hepatic Markers at Rejection

Table 2 shows the basal time point of BPR (R) and 1–2 days before BPR (Pre-R) values of ds-cfDNA percentage and functional liver marker values at rejection: alanine aminotransferase (ALT), aspartate aminotransferase (AST), gamma-glutamyltransferase (gamma-GT), and bilirubin. As ALT, AST, gamma-GT, and bilirubin values were considerably high in many patients during the first month after TX, we included the values obtained 2 years after TX in the statistical analysis. ANOVA showed significant differences for both; levels of ds-cfDNA and liver function markers increased, except for ALT values; however, discrimination between basal, Pre-R, and R groups was more significant for graft cfDNA than for liver marker levels.

The percentage increase in the basal levels of ds-cfDNA at R and Pre-R is shown in Figure 3. Post hoc analysis showed a significant increase at rejection compared with basal values (*p* < 0.0001) and Pre-R values (*p* < 0.05). This significant increase over basal levels was observed not only at the point of BPR but also 2 days prior (*p* = 0.001).

Regarding hepatic markers, Figure 4 shows great data dispersion and higher overlap between the basal, R, and Pre-R groups. Post hoc analysis of gamma-GT values showed significant differences between the basal and R values (*p* < 0.01). When compared with values obtained after 2 years, we observed significant differences between this value and either basal (*p* < 0.01), pre-R, or R values (*p* < 0.0001). Similarly, the bilirubin level post hoc analysis showed differences between basal and R values (*p* = 0.005), 2-year and either Pre-R (*p* < 0.05) or R values (*p* < 0.0001), and Pre-R and R values (*p* < 0.05). AST values showed only significant differences between the 2-year and either pre-R or R values (*p* < 0.05). Total cfDNA measured by amplification of the beta-globin gene was also compared at basal, R, and Pre-R; high basal levels were detected, and no significant differences were observed by ANOVA analysis (Table 2).

Both the total population and stable group of patients showed a significant correlation between functional liver biomarker levels and ds-cfDNA percentage (Spearman test; Table 3). Patients with BPR during the first month also showed a significant correlation, except for the gamma-GT values. However, although *p*-values were clearly significant for most analyses, Spearman correlation coefficients were not always close to one.

Receiver operator characteristic (ROC) analysis was performed to determine the ability of both graft cfDNA and hepatic markers to discriminate acute rejection from stable samples. Samples included in the analysis were of the patients with BPR during the first month of follow-up, which were obtained in a period of 4 days before or after the R. Baseline samples come from stable patients during the first month of follow-up after patient stabilization, and it consisted of decreased ds-cfDNA values at 10–15 days after TX. Finally, 100 cases were included, of which 16 samples came from patients with BPR during the first month of follow-up (7 cases were lost in the model).

The ROC analysis data are summarized in Table 4. When compared with classical liver function markers, graft cfDNA with high AUC (AUC = 96.5; 95% CI = 91.2–101.2) is clearly better at discriminating rejection from stable samples. Among them, gamma-GT showed the best AUC, and in contrast, ALT values presented a poor discrimination with a 95% CI < 50 (AUC = 76.1; 95% CI = 62.7–89.5; AUC = 60.3; 95% CI = 43–77.6 respectively). A threshold cut-off value of 8.6% for graft cfDNA increase was determined from simultaneous maximization of sensitivity (93.8; 95% CI = 72–99) and specificity (0.87.8; 95% CI = 79–93). Positive (PPV) and negative predictive value (NPV) calculated using the diagnostic threshold of 8.6% for ds-cfDNA increase were 60% (95% CI = 41–77) and 99 (95% CI = 93–100), respectively. Figure 5 illustrates the ROC curve for all parameters studied.

## 4. Discussion

Graft cfDNA released from damaged donor liver cells in the circulation of liver transplant recipients has been proposed as a potential noninvasive biomarker of organ rejection. In this study, transplanted liver rejection was evaluated using real-time PCR. We observed a significant increase in ds-cfDNA over basal levels at the BPR time point and several days before rejection.

In recent years, several authors have proposed the quantification of ds-cfDNA in the circulation of transplant recipients as a potential biomarker for organ rejection or cellular graft injury [4,15,16]. The direct assessment of liver integrity could lead to an earlier detection of acute rejection, which could provide prompt intervention and facilitate the adjustment of an effective therapy. Different techniques have been used for monitoring ds-cfDNA in the host serum, such as analyses of single-nucleotide polymorphisms (SNP), which discriminate between heterologous alleles in the recipient and the donor by means of digital PCR [7]. Other methods include next-generation sequencing (NGS) analysis [4,5,6] and evaluation of insertion/deletion polymorphisms [8,9,12]. We developed a qPCR method for the quantification of serum ds-cfDNA in liver transplant patients based on the amplification of a panel of several InDels present in the donor but not in the recipient. Other research groups have analyzed InDel polymorphisms commonly found in the general population for the detection of ds-cfDNA based on digital PCR quantification [9] or qPCR [12]. As we propose herewith, Adamek et al. also designed primers and probes inside the insertion or deletion sequence for amplifying by qPCR only when the insertion is present [12]. However, they included a first step for genotyping donor and recipient by a PCR followed by gel electrophoresis in order to distinguish the InDel polymorphisms by the amplicon size. We avoided the step of gel electrophoresis analysis, simplifying the procedure. On the other hand, a digital PCR approach may be very useful when the amount of ds-cfDNA after graft damage is too low and a more sensitive technique is needed [8,9]. However, for liver transplantation qPCR has been proven to be a suitable technique capable of detecting ds-cfDNA in the recipient cfDNA. The proposed InDel panel allowed the monitoring of most of the patients from our population after TX, although the frequency and informativeness of the different InDels were variable. To avoid the limitation of a lack of an informative donor–recipient mismatch for some patients, we considered adding further loci to the genotyping panel. Moreover, although in our area it is not common to receive a liver from a related donor, in this case, the probability of a lack of an informative locus increases. Thus, to fully extend the monitoring to the entire population, we expanded the panel to detect a higher number of polymorphic InDels. For all included InDels, we observed an early increase during the first 24 h after TX, with a different decay after this time point, depending on the clinical evolution of the patient.

A small number of patients experienced transient acute rejection during the first month after TX. As previously described, although there was an elevation of InDel percentage over basal levels at the time point of BPR, values returned to baseline after successful treatment. Since ds-cfDNA is quickly cleared from the serum, it is considered an acute specific biomarker, in contrast to the markers that gradually decrease over a long period, as seen with liver function markers such as aminotransferases enzymes. We observed significant differences in ds-cfDNA increase over basal levels, not only at the BPR time point but also several days before that. Other studies using NGS or SNP analysis [4,16] also observed an increase in ds-cfDNA prior to rejection diagnosis. Differences were also found when functional liver markers were analyzed (ALT, AST, gamma-GT, and bilirubin), although discrimination between the basal, Pre-R, and R groups was clearly more significant for graft cfDNA. Most patients with rejection during the first month of follow-up showed an elevation in serum liver biomarker levels over baseline levels. However, this elevation was also observed in many patients with good evolution, and in most cases, the liver marker levels did not decrease to basal levels until several months after TX. Thus, in the ANOVA comparative analysis of samples under or close to BPR and basal samples, we included the vales obtained 2 years after TX to achieve stable basal levels. In fact, only gamma-GT and bilirubin values showed significant differences between basal and BPR sample values.

Our results showed that ds-cfDNA levels discriminated BPR with an AUC of 96.5 obtained from the ROC analysis. Compared with functional liver markers, the results of ROC analysis indicated a higher diagnostic sensitivity for ds-cfDNA percentage. Maximizing both sensitivity and specificity, a threshold cutoff of 8.6% provided an estimated NPV of 99% and a PPV of 60%. The predictive values of rejection for defined thresholds have been well established in several reports regarding kidney and heart transplants. Thus, the pooled NPV described from studies in kidney and heart transplant patients ranged from 75% to 98%, and the combined PPV reported was rather low, ranging from 12% to 77% [6,17].

Few reports have analyzed the predictive values of liver TX [18,19]. The NPV values were always approximately 100%, similar to the value obtained in our study (99%; 95% CI = 93–100). We believe that the NPV values obtained, together with those previously published, are clinically promising. Thus, the high NPV suggests that this test may be helpful in avoiding unnecessary biopsies only triggered by elevated liver biomarker values.

The PPV reported for liver BPR was in the average range of 55%–62.5%. The data presented showed a PPV within this range (60%; 95% CI = 41–77). These relatively low PPV values may be explained by two facts: the relatively low prevalence of rejection for liver TX, and because the ds-cfDNA quantification is not a specific marker of rejection, it may be associated with general liver damage. In this way, the PPV obtained suggest that ds-cfDNA may be an adjuvant value that, together with other clinical evidences of rejection, provides valuable information about the liver health in order to make the decision to perform the biopsy.

Individually designed treatment is an important issue that must be addressed. In this context, individualized immunosuppressant dose evaluations using ds-cfDNA detection should be considered. Thus, monitoring ds-cfDNA may be a useful tool for patient stratification, which may allow for selection of patients with higher levels of ds-cfDNA and probably greater rejection risk, which may require more intensive therapy. In addition, during immunosuppressant drug minimization attempts [20], ds-cfDNA levels may guide the dose reduction protocol. Moreover, ds-cfDNA quantification could help in assessing the efficacy of drug dose changes or testing a possible underdose of immunosuppressant drug due to drug toxicity or poor adherence to the treatment. Thus, another advantage of ds-cfDNA quantification is its potential to identify patients with low immunosuppression levels. It could be shown that in a subgroup of patients with tacrolimus concentrations below the therapeutic range, ds-cfDNA may be significantly increased.

The different approaches used to quantify ds-cfDNA in the recipient circulation have some limitations that might reduce their clinical effectiveness. The goal of testing is to improve graft survival, but it must have a reasonable turnaround time and must be available at a reasonable cost. The test described here involves a rapid turnaround process that may allow for the adoption of decisions and modifications in the clinical management of patients after TX. One limitation of this approach may be the low amount of ds-cfDNA that is usually diluted in the recipient cfDNA. Thus, different organs may release different amounts of cfDNA [21]. Although we were able to quantify ds-cfDNA in liver transplant patients by qPCR [11], a more sensitive technique such as digital PCR is needed in heart transplant patients, as reported in a previous study [8]. In addition, the variable fragmentation of cfDNA should be considered in order to design the optimal PCR conditions for amplification [22]. Finally, increases in ds-cfDNA may be the consequence of other liver damage in addition to rejection, including immunosuppressive drug toxicity, post-operative complications, or recurrence of primary disease. Therefore, ds-cfDNA in recipient’s serum by itself may not be a definitive diagnostic test for identification of rejection and should be considered as an adjunct rather than a definitive tool to replace biopsy.

The results from this prospective study validated and extended prior reports suggesting that ds-cfDNA measured in liver transplant patients may be a useful marker of graft integrity, helping to identify patients with acute rejection better than other conventional liver biomarkers. The negative predictive value obtained in this study confirms that ds-cfDNA quantification using qPCR may be a helpful adjuvant test to prevent rejection, avoiding unnecessary biopsies due to elevated liver biomarker values. Further research should be conducted to increase the number of patients to validate the proposed approach. Moreover, an extended panel would allow us to detect a higher number of polymorphic InDels.

## 5. Conclusions

We described a significant increase in ds-cfDNA related to liver damage not only at BPR but also several days before. Results obtained from ROC analysis showed that ds-cfDNA levels discriminated BPR with a higher diagnostic sensitivity compared with functional liver markers. Moreover, the high negative predictive value obtained in this study suggest that ds-cfDNA quantification using qPCR may be helpful in avoiding unnecessary biopsies when an increase in liver biomarkers is observed in the absence of ds-cfDNA elevation. Increases in ds-cfDNA in liver transplanted patients, however, may not be considered a definitive rejection diagnostic test replacing biopsy and may reflect the liver damage and provide valuable information in order to either make the decision to perform the biopsy or change the immunosuppressive treatment with the intention of normalizing the altered biomarker levels.

## Figures and Tables

**Figure 1 jcm-12-00036-f001:**
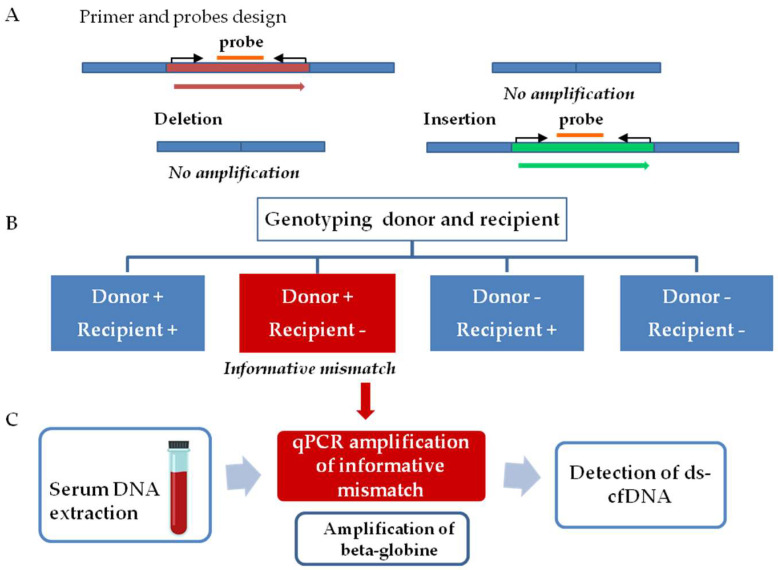
Workflow for detection of ds-cfDNA. (**A**) Primers and probes were designed for amplification of small fragments (<150 pb) on the insertion/deletion sequence. The workflow involves two steps. (**B**) Genotyping donor and recipient samples. A mismatch was considered informative when a deleted sequence on the host DNA but not in the donor DNA. (**C**) cfDNA extraction from sequential serum samples and amplification by qPCR of the informative mismatch and beta-globin gene. The amplification of the informative mismatch during the follow up involves the presence of ds-cfDNA.

**Figure 2 jcm-12-00036-f002:**
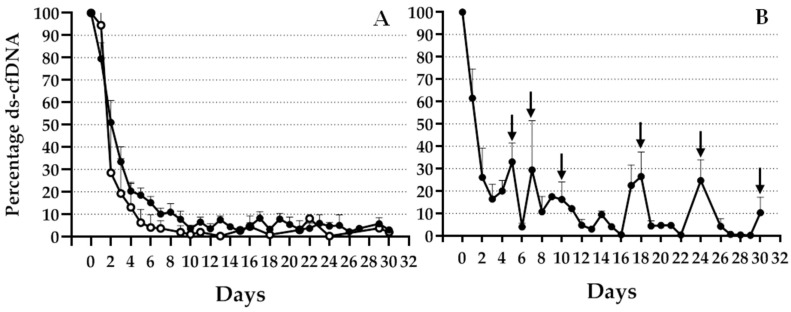
Percentage decrease in ds-cfDNA during the first month post TX. Percentage decrease in ds-cfDNA levels (KEG/mL) from day 0 of TX. (**A**) Patients who accepted transplanted livers without any complications and rejection (black circles) or rejection during a 2-year period (white circles). (**B**) Patients with BPR during the first month after TX. Data expressed as mean + SEM.

**Figure 3 jcm-12-00036-f003:**
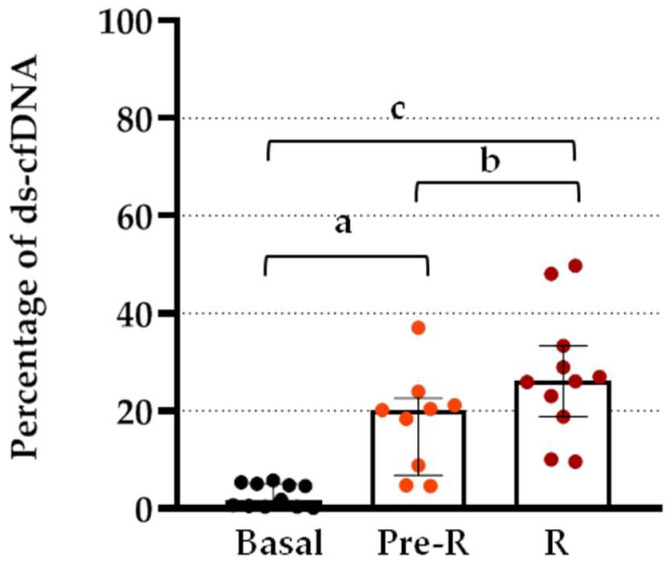
Differences in serum ds-cfDNA levels at BPR and previous to BPR from patients after TX. Percentage increase in serum ds-cfDNA in patients after TX at BPR (R) and 2–3 days before BPR (pre-R) compared to basal levels. Boxes represent median and inter-quartile ranges, and whiskers show 5th–95th percentile. ANOVA test *p* < 0.0001. Analysis between groups by post hoc Bonferroni test; (a) *p* = 0.001; (b) *p* < 0.05; (c) *p* < 0.0001.

**Figure 4 jcm-12-00036-f004:**
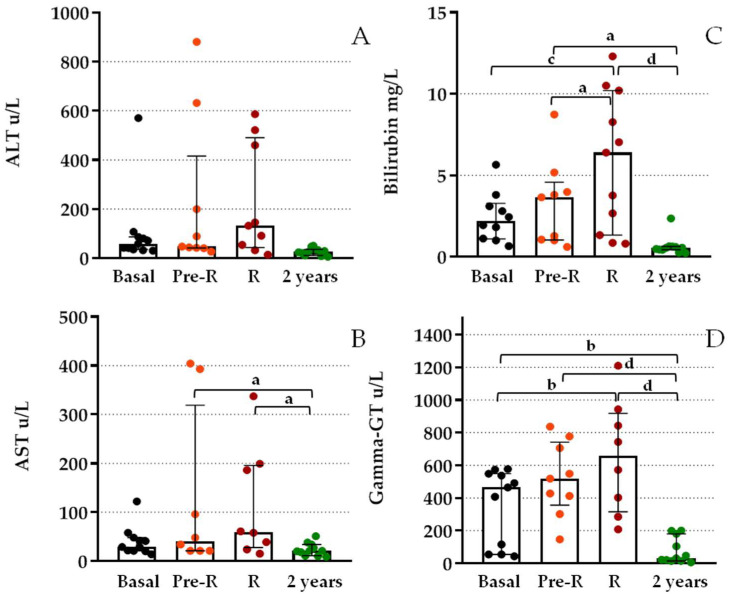
Differences in functional liver marker levels at BPR and previous to BPR from patients after TX. (**A**) Serum alanine aminotransferase (ALT); (**B**) aspartate aminotransferase (AST); (**C**) bilirubin; and (**D**) gamma-glutamyltransferase (gamma-GT) levels of patients after TX at BPR (R), 2–3 days before BPR (pre-R) compared to basal levels and 2 years after TX. Boxes represent median and inter-quartile ranges and whiskers show 5th–95th percentile. ANOVA test *p* < 0.0001. Analysis between groups by post hoc Bonferroni test; (a) *p* < 0.05; (b) *p* < 0.01; (c) *p* = 0.005; (d) *p* < 0.0001.

**Figure 5 jcm-12-00036-f005:**
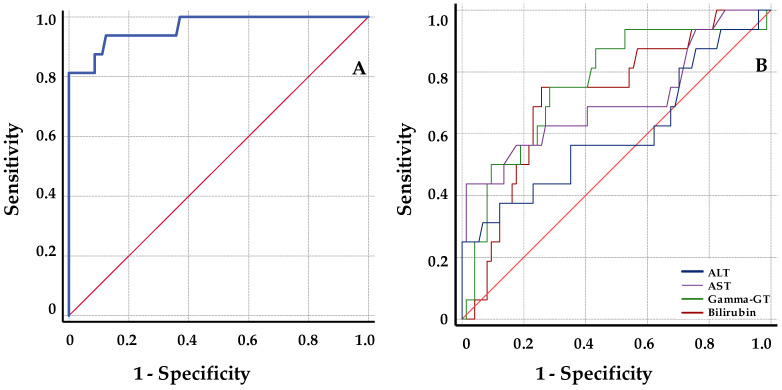
ROC curves for discrimination of rejection from stable samples. (**A**) ROC curve for serum ds-cfDNA percentage values; (**B**) ROC curves for serum liver function markers. Of the 100 samples included, 84 were baseline samples and 16 were samples of patients under BPR. ALT: alanine aminotransferase; AST: aspartate aminotransferase; gamma-GT: gamma-glutamyltransferase.

**Table 1 jcm-12-00036-t001:** General characteristic of the patients.

	Total (*n* = 97)	Stable (*n* = 84)	Rejection (*n* = 13)	*p*
Age (years)	55 [18–71]	56 [18–71]	49 [21–60]	**0.0003**
Sex (male)	50 (51.5%)	42 (50%)	8 (61.5%)	0.55
Warm ischemia (min)	30 [12.5–60]	30 [12.5–60]	30 [22–36]	0.54
Cold ischemia (min)	382 [250–551]	382 [270–551]	360 [250–425]	0.16
Previous ICU	10 (10.3%)	7 (8.3%)	3 (23.1%)	0.13
Previous ICU (days)	3 [1–58]	3 [1–58]	1 [1–34]	0.42
ICU post-TX (days)	6 [3–54]	6 [3–54]	6 [4–14]	0.768
Hospitalization (days)	18 [4–109]	17 [4–109]	30 [11–67]	**0.01**
Beta-globin 48 h (GKE/mL)	321 [104–4979]	346 [104–3232]	270 [104–4979]	0.19
Rejection during first month	13 (13.4%)			
Rejection (two years after TX)	20 (20.6%)			

Continuous values expressed as median [range]. Beta-globin values are the mean of 48 h values after TX; GKE: genomic kilo equivalent. Mann–Whitney U-test was used to compare differences for all continuous parameters studied with exception of cold ischemia that was normally distributed (*t*-test). Bold: significant *p* values (*p* < 0.05).

**Table 2 jcm-12-00036-t002:** Comparison of values of percentage of graft cfDNA and classical hepatic markers at basal, 2 years after TX, BPR, and 2 days before BPR time points.

Mean + SEM	Basal	Pre-R	R	2 Years	*p*
ds-cfDNA	2.73 + 0.75	17.64 + 3.44	29.27+ 4.32	-	**<0.0001**
cfDNA	60.07 + 7.17	85.17 + 9.23	70.08 + 4.15	-	0.061
ALT	104.81 + 47.13	222.22 + 104.53	226.11 + 76.11	26.0 9 + 4.47	0.089
AST	40.54 + 9.1	129.75 + 59.3	114.87 + 40.44	23.63 + 3.92	**0.047**
Gamma-GT	351.45 + 69.86	519.22 + 75.39	650.87 + 122.54	76.09 + 24.04	**<0.0001**
Bilirubin	2.43 + 0.47	3.25 + 0.87	5.82 + 1.26	0.65 + 0.17	**0.001**

Ds-cfDNA and cfDNA expressed as percentage of maximal value at TX day 0. Multiple comparison analysis by ANOVA test. R: BPR sample; Pre-R 1–2 days previous BPR; ALT: alanine aminotransferase; AST: aspartate aminotransferase; Gamma-GT: gamma-glutamyltransferase. Bold: significant *p* values (*p* < 0.05).

**Table 3 jcm-12-00036-t003:** Correlation between ds-cfDNA percentage and liver function markers for total population after TX, stable patients, and patients suffering BPR.

ds-cfDNA	Total		Stable Patients		Patients with BPR	
	CC	*p*	CC	*p*	CC	*p*
ALT	0.58 **	**<0.0001**	0.61 **	**<0.0001**	0.43 **	**0.0001**
AST	0.66 **	**<0.0001**	0.72 **	**<0.0001**	0.50 **	**<0.0001**
Gamma-GT	−0.26 **	**<0.0001**	−0.32 **	**<0.0001**	−0.05	0.72
Bilirubin	0.2 **	**<0.0001**	0.16 *	**0.01**	0.38 **	**0.001**

Total (*n* = 347); stable patients (*n* = 258); BPR patients (*n* = 89). Ds-cfDNA expressed as percentage of maximal value at TX day 0. Spearman test correlation coefficient (CC) with 95% (*) or 99% (**) CI; significant *p* value in bold. ALT: alanine aminotransferase; AST: aspartate aminotransferase; Gamma-GT: gamma-glutamyltransferase.

**Table 4 jcm-12-00036-t004:** Youden index diagnostic sensitivity and specificity from receiver operator characteristic (ROC) curve in rejection versus stable samples.

Determination	ÁUC (%)	Signification	95% CI	Sensitivity (%)	Specificity (%)	Threshold Value at Maximum YI
ds-cfDNA	96.5	<0.0001	91.9–101.2	93.75	87.84	8.605 (%)
ALT	60.3	0.242	43–77.6	37.5	87.84	144.5
AST	70.8	0.012	54.5–87.1	43.75	98.7	94.5
Gamma-GT	76.1	0.0001	62.7–89.5	75	71.7	401.5
Bilirubin	72.1	0.001	58.7–85.4	75	74.3	2.65

AUC: area under the curve; YI: Youden index. Ds-cfDNA expressed as percentage of maximal value at TX day 0. ALT: alanine aminotransferase; AST: aspartate aminotransferase; Gamma-GT: gamma-glutamyltransferase.

## Data Availability

The datasets generated and analyzed during the current study are available from the corresponding author upon reasonable request. The data are not publicly available due to ethical reasons.

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
