# Peer review of "Donor-Specific Cell-Free DNA qPCR Quantification as a Noninvasive Accurate Biomarker for Early Rejection Detection in Liver Transplantation"

_jcm, 2022, doi:10.3390/jcm12010036_

Round 1

Reviewer 1 Report

There are several commercially available markers for cf-DNA testing to screen for rejection in transplant patients, particularly using next generation sequencing. So the essential merit of the study is in the attempt to standardize a real time PCR based screening tool for rejection in liver transplant patients. 

However, the major demerit of the study is in the presentation of donor specific cf-DNA in recipient's serum as a definitive diagnostic test for identification of rejection. Also, the authors have repeatedly stated that the aim of their study is to avoid biopsies. These statements are in conflict with their own findings that the episodes of rejection in liver transplant patients are rarer and the cf-DNA test had a low positive predictive value. Biopsy as a test is essentially a specific test, with a high positive predictive value for definitive diagnosis while the cf-DNA test is a screening investigation with high sensitivity and negative predictive value.

The study definitely proves that circulating donor cf-DNA is a sign of damage of hepatocytes and there is a temporal association. However, the damage to hepatocytes are caused by several insults including drug toxicity from immunosuppression, hypoxic insults, free-oxide injury, infections, nutrition in immediate post-operative period, recurrence of primary disease etc. in addition to rejection. 

Though the scientific methodology is acceptable, the interpretation and the presentation of their results as a tool to replace biopsy with cf-DNA, rather than as an adjunct, conveys wrong message.  

Author Response

There are several commercially available markers for cf-DNA testing to screen for rejection in transplant patients, particularly using next generation sequencing. So the essential merit of the study is in the attempt to standardize a real time PCR based screening tool for rejection in liver transplant patients. 

However, the major demerit of the study is in the presentation of donor specific cf-DNA in recipient's serum as a definitive diagnostic test for identification of rejection. Also, the authors have repeatedly stated that the aim of their study is to avoid biopsies. These statements are in conflict with their own findings that the episodes of rejection in liver transplant patients are rarer and the cf-DNA test had a low positive predictive value. Biopsy as a test is essentially a specific test, with a high positive predictive value for definitive diagnosis while the cf-DNA test is a screening investigation with high sensitivity and negative predictive value.

The study definitely proves that circulating donor cf-DNA is a sign of damage of hepatocytes and there is a temporal association. However, the damage to hepatocytes are caused by several insults including drug toxicity from immunosuppression, hypoxic insults, free-oxide injury, infections, nutrition in immediate post-operative period, recurrence of primary disease etc. in addition to rejection. 

Though the scientific methodology is acceptable, the interpretation and the presentation of their results as a tool to replace biopsy with cf-DNA, rather than as an adjunct, conveys wrong message.  

Thank you very much for your suggestions that will improve the manuscript quality. We tried to modify the discussion section and conclusions with a more accurate interpretation of the results.

We agree that, from the results obtained, we cannot conclude that ds-cfDNA in recipient's serum is a definitive diagnostic test for identification of rejection, and other liver damage should be considered.  We have modified the discussion and conclusion according with the referee suggestions (in red). Page 2, final paragraph of Introduction; Discussion section, page 12, last paragraph and page 13, first paragraph; Conclusions, most of the conclusion paragraph.

In relation of the positive and negative predictive value (PPV, NPV) obtained, we probably did not clearly explain our interpretation of the data. We mean that high NPV suggest that, when an increase of liver biomarkers is observed in the absence of ds-cfDNA elevation liver biopsy may be avoided. The PPV obtained suggest that ds-cfDNA may be an adjuvant value that, together with the elevation of liver markers, leads the clinician to make the decision to perform the biopsy, or at least to change the immunosuppressive treatment with the intention of normalizing both biomarkers levels. We have clarified this issue on the Discussion section (in red). Page 12, third paragraph.

Reviewer 2 Report

This manuscript described a significant increase in ds-cfDNA related to liver damage not only at BPR but also several days before. Results obtained from ROC analysis showed that ds-cfDNA levels discriminated BPR with a higher diagnostic sensitivity compared with functional liver markers. The topic is interesting and has provided a practical method for monitoring the status of organ transplantation. However, there are a few points that should be addressed.

(1) Similar research [13] has been published previously, please specify the advantages and disadvantages of these two reported methods in the Discussion section.

(2) For the definition of ds-cfDNA, it is desirable to have a schematic diagram.

(3) How to calculate the proportion of ds-cfDNA, it is desirable to provide a formula.

(4) If the donor and the recipient are related (which should be common, such as father and son, siblings, etc.), will the probability of a lack of an informative locus between them increase? The number of genetic markers required will be further increased correspondingly, so can a sufficient number of genetic markers be detected in a single run by the q-PCR method? Please emphasize these in the Discussion section.

Author Response

This manuscript described a significant increase in ds-cfDNA related to liver damage not only at BPR but also several days before. Results obtained from ROC analysis showed that ds-cfDNA levels discriminated BPR with a higher diagnostic sensitivity compared with functional liver markers. The topic is interesting and has provided a practical method for monitoring the status of organ transplantation. However, there are a few points that should be addressed.

Thank you very much for your suggestions that would improve the manuscript quality. We tried to clarify the points required

(1) Similar research [13] has been published previously, please specify the advantages and disadvantages of these two reported methods in the Discussion section.

As far as I know, two groups have proposed a similar approach (ref 9 and 12). They analyze InDel polymorphisms commonly found in the general population for the detection of ds-cfDNA, although different designs have been carried out. Both scientific methodologies are valuables, but in our hands, the test described in the present study involves a rapid turnaround process and it is based on qPCR assay, a technology available in most of the clinical biochemistry laboratory.

Goh and col. first genotype donor and recipient InDel by qPCR analyzing the presence or absence of the insertion by the amplicom size by differences in the high-resolution melting analysis. In a second step, they design a digital PCR whereas only the donor specific cfDNA is amplified. We consider that for some organ transplantation, as reported for heart transplant, the amount of ds-cfDNA after graft damage may be too low, and a more sensitive technique, such as digital PCR, would be needed. However, for liver transplantation qPCR has been probe to be a suitable technique capable to detect ds-cfDNA in the recipient cfDNA.

Adamek and col. have reported a method more similar to the one proposed by our group, as they quantified ds-cfDNA during the follow up by qPCR. As we report, they also design primers and probes inside the insertion or deletion sequence for amplifying only when the insertion is present. However, they first genotype donor and recipient by a PCR followed of gel electrophoresis to analyzed amplicon size in order to distinguish the InDel polymorphisms. We think that the same methodology may be used for both genotyping and monitoring ds-cfDNA avoiding the step of gel electrophoresis analysis.

 We have included this explanation in the Discussion section (page 11, second paragraph).

 (2) For the definition of ds-cfDNA, it is desirable to have a schematic diagram.

We have included a schematic diagram explaining the workflow for ds-cfDNA determination (figure 1)

 (3) How to calculate the proportion of ds-cfDNA, it is desirable to provide a formula.

We do not calculate the proportion of ds-cfDNA from total cfDNA. We quantify the ds-cfDNA that diminishes after the first days following TX (figure 2). Any increase over this basal level should be considered a sign of graft damage. The formula for calculating final concentration of cfDNA /mL serum is detailed on the methods section (page 5)

(4) If the donor and the recipient are related (which should be common, such as father and son, siblings, etc.), will the probability of a lack of an informative locus between them increase? The number of genetic markers required will be further increased correspondingly, so can a sufficient number of genetic markers be detected in a single run by the q-PCR method? Please emphasize these in the Discussion section.

All patients included received a liver for a death donor. In our area it is not common to receive a liver from a related donor, as usually occurs with kidney transplant. As we comment in the discussion section we are now expanding the number of loci to the genotyping panel. Thus, we could detect a higher number of polymorphic InDels and fully extend the monitoring to the entire population. Anyway, we comment this issue in the discussion section to remark this possible limitation (page 11, second paragraph)

Round 2

Reviewer 1 Report

Significant changes have been made to the manuscript. 

Minor corrections to be considered are 

1. Conclusions in Abstract - dscfDNA  may be a useful marker for graft integrity in liver transplant patients to prevent rejection and avoid unnecessary biopsies. The statement "prevent rejection and avoid unnecessary biopsies" may be modified as to "screen for rejection and liver damage"

2. In Introduction, Page 2, line 75-76 -  the statement "be helpful in avoiding unnecessary biopsies triggered by elevated liver biomarker values" may be modified as "be helpful in deciding on the indication for biopsy in the setting of elevated liver biomarkers." The words unnecessary biopsies conveys a wrong interpretation the scientific finding in the study and may be avoided.

Author Response

Thank you very much for the last corrections to the manuscript proposed.

  1. Conclusions in Abstract - dscfDNA  may be a useful marker for graft integrity in liver transplant patients to prevent rejection and avoid unnecessary biopsies. The statement "prevent rejection and avoid unnecessary biopsies" may be modified as to "screen for rejection and liver damage"

We have changed the sentence of the Abstract conclusion as required (in red).

  1. In Introduction, Page 2, line 75-76 -  the statement "be helpful in avoiding unnecessary biopsies triggered by elevated liver biomarker values" may be modified as "be helpful in deciding on the indication for biopsy in the setting of elevated liver biomarkers." The words unnecessary biopsies conveys a wrong interpretation the scientific finding in the study and may be avoided.

We have changed the sentence at the end of the Introduction as required (in red).